# A Bench-Top Approach for Isolation of Single Antibody Producing Chinese Hamster Ovary (CHO) Cells Using a Microwell-Based Microfluidic Device

**DOI:** 10.3390/mi13111939

**Published:** 2022-11-10

**Authors:** Salma Fuadiyah, Kittipat Chotchindakun, Rungrueang Phatthanakun, Panwong Kuntanawat, Montarop Yamabhai

**Affiliations:** 1School of Biotechnology, Institute of Agricultural Technology, Suranaree University of Technology, Nakhon Ratchasima 30000, Thailand; 2Institute of Research and Development, Suranaree University of Technology, Nakhon Ratchasima 30000, Thailand; 3Synchrotron Light Research Institute, Nakhon Ratchasima 30000, Thailand

**Keywords:** monoclonal cell lines, cell line generation, single-cell cloning, limiting dilution, microfluidic device

## Abstract

Genetically-modified monoclonal cell lines are currently used for monoclonal antibody (mAbs) production and drug development. The isolation of single transformed cells is the main hindrance in the generation of monoclonal lines. Although the conventional limiting dilution method is time-consuming, laborious, and skill-intensive, high-end approaches such as fluorescence-activated cell sorting (FACS) are less accessible to general laboratories. Here, we report a bench-top approach for isolating single Chinese hamster ovary (CHO) cells using an adapted version of a simple microwell-based microfluidic (MBM) device previously reported by our group. After loading the cell suspension to the device, the electrostatically trapped cells can be viewed under a microscope and transferred using a micropipette for further clone establishment. Compared to the conventional method, the invented approach provided a 4.7-fold increase in the number of single cells isolated per round of cell loading and demonstrated a 1.9-fold decrease in total performing time. Additionally, the percentage of correct single-cell identifications was significantly improved, especially in novice testers, suggesting a reduced skill barrier in performing the task. This novel approach could serve as a simple, affordable, efficient, and less skill-intensive alternative to the conventional single-cell isolation for monoclonal cell line establishment.

## 1. Introduction

Monoclonal antibodies (mAbs) are monovalent-affinity-specific antigen-binding antibodies and are considered promising candidates in novel drug discovery. Moreover, mAbs can serve as therapeutic agents to treat several serious diseases such as cancer, inflammatory, cardiovascular, and autoimmune diseases [1,2]. Due to their pharmaceutical advantages and global demand, the market value of mAbs was valued at about US$115.2 billion in 2018 and is expected to increase to $300 billion by 2025 [2].

Over the past decades, several technologies have been developed for commercial mAbs production. Hybridoma, first developed in 1975, is a traditional technology to generate highly specific mAbs from a provoked animal immune response [2]. However, mAbs produced using this technology are not stable, and the technique is not reproducible, hindering large-scale production with this method [3]. Most recently, genetic engineering has been used to fix this drawback by constructing a stable high-producing clone for mAbs [4].

Among various host cells, mammalian cell lines such as Chinese Hamster Ovary (CHO) cells have been considered suitable for mAbs production due to their high growth rate in suspension form and their ability to secrete highly complex biomolecules with a glycosylation profile similar to human ones [5]. It is the most used mammalian cell line for commercial mAbs production [6].

Since mAbs production by CHO cells can be obtained by randomly transfecting a DNA plasmid containing the gene of interest into the cells, the obtained transfected cell populations usually show diverse gene expression [7,8]. Therefore, single-cell isolation is required for separating cells individually, thereby screening and specifically selecting the targeted high-producing and stable mAbs-expressing clones.

Limiting dilution is a conventional method of single-cell isolation based on Poisson distribution, achieving single-cell production from diluted cell suspensions using pipetting into 96-well plates. However, its low efficiency and the high repetition required are the main hindrances of this method, generating low monoclonality and time-consuming procedures [9]. Although Fluorescence-Activated Cell Sorting (FACS), another high-throughput technique with fluorescence sorting, which shows high efficiency in single-cell isolation, is less laborious and less time-consuming [10], this technique requires expensive reagents, skillful operating, and is prone to cell damage due to the hydrodynamic stress applied during the sorting process [11,12]. Consequently, new single-cell isolation techniques that can provide high efficiency with less effort and decreased operating time are in demand and being developed.

For instance, microfluidic systems have emerged as valuable tools for cell manipulation in mAbs production over the past decades, especially in single-cell isolation processes, due to their great potential for functional high-throughput screening at a single-cell level. Microfluidic devices can indeed increase the efficiency of single-cell isolation techniques, require fewer operating skills, and also economize reagent usage. However, their structural complexity might be the main barrier to their fabrication [13,14]. In addition, either specialized visualization, optical equipment, or automation, at a minimum, may be a prerequisite to perform the task [13,14,15]. It could be difficult for a general laboratory to benefit from these advancements. Hence, the development of a simple-structural microfluidic device that requires less or no advanced operation is critical to overcoming such limitations in order to improve mAbs production.

The present study aims to implement a simplified version of the existing bench-top microwell-based microfluidic (MBM) device from Kuntanawat and colleagues (2014) for isolating high-mAbs-producing CHO cells. We use CHO cells as a model organism in this study because they have been considered suitable for high mAbs production. The isolation mechanism of the MBM device is based on the electrostatic trapping of cells and with the simple use of the force of gravity. This MBM device has previously been demonstrated to successfully isolate multiple microalgal cell types, such as *Spirulina platensis*, *Chlorella vulgaris*, and *Scenedesmus* spp., and there is a clear potential for this device to be modified and applied in mammalian cell linage development. The modification of the device from the existing solution allows for simple fabrication of the microstructure using a laser cutting machine and less technical device assembly. The device allows trapped single cells to be grown individually for morphological development, assessment, and growth tracking. Each of them later develops into a single monoclone, which can easily be transferred using micropipetting. In this work, the isolation parameters, including the cell loading concentration and the settling time, were optimized to enhance the efficiency of single-CHO cell isolation using the MBM device. Moreover, we evaluated the total time-spent processing on the MBM device compared to a conventional technique, i.e., the limiting dilution method by volunteer investigation. It was hypothesized that the greater efficiency of single-CHO cell isolation obtained from the MBM device, the less the total time-spent processing that its remarkable efficiency and simple operation could accomplish.

## 2. Materials and Methods

### 2.1. Design and Fabrication of MBM Device

The MBM device was modified corresponding to our previous study [16], in which the upper cover and the inlet and outlet channels were removed. The device is composed of three components, including a fluidic layer, a microwell layer, and a glass slide layer. The pattern of the device is presented in Figure 1. Poly methyl methacrylate was used as a material for a fluidic layer and a microwell layer, fabricated using a laser cutting technique (HanMa Laser model HM-1310J, Guangzhou, China). A microwell layer contained a series of 200 wells comprising 10 rows of 20 wells. Each well had a dimension (width × length × depth) of 1000 µm × 1000 µm × 1000 µm, with a spacing of 500 µm. To assemble the device, the fluidic layer and a microwell layer were aligned and attached to a glass slide layer, i.e., a positive charge microscope slide (75 mm × 25 mm × 1 mm) (Superfrost^®^ Plus J1800AMNZ, Saarbrückene, Germany). Afterward, a silicone mixture (Silicone SF 820 A and Silicone SF 820 B) was used as an adhesive agent. All components were allowed to fix for 15 min completely. An experiment of cell trapping using a similar device assembled with an ordinary glass slide was found to neither efficiently trap cells nor prevent cells from escaping the wells [16]. This is the reason why this current MBM device was assembled using the positively charged glass slide.

### 2.2. Chinese Hamster Ovary Cell Culture and Harvesting

Chinese Hamster Ovary (CHO) cell lines in this study were obtained from Molecular Biotechnology Laboratory (MY Lab) of SUT [17]. Cell cultivation and maintenance were performed as previously described [18]. HyClone ActiPro medium (Cat: HAD3103701, Marlborough, MA, USA), 8 mM glutamine (Cat: 25030–081), and 0.2% anti-clumping (Cat: 0010057DG) were supplemented to the cell culture media. Cells were cultivated in 125 mL Erlenmeyer flasks with a working volume of 20 mL in a shaking incubator at 150 rpm, 37 °C, and 7% carbon dioxide. The passaging of cells was performed regularly to maintain cells in the exponential phase (approximately 3 × 10^5^ cells mL^−1^). For harvesting, cells were centrifuged (Biosan, LCM-3000, Riga, Latvia) at 1000 RPM for 5 min and resuspended in sterile phosphate buffer saline (PBS) at pH 7.4.

### 2.3. Operation of MBM Device for Single-Cell Isolation

Prior to isolating a single cell, the device was sterilized by rinsing with 70% (*v*/*v*) ethanol and sterile deionized water, respectively. The operation of the device is depicted in Figure 2, divided into three parts, including cell preparation, cell loading, and single-cell assessment. Briefly, the microwell and fluidic layers were filled to maximum capacity with 3 mL of cell culture medium. Next, the CHO cell suspension was prepared to approximate a cell concentration of 10 × 10^3^ cell mL^−1^. Subsequently, 1 mL of cell culture medium in a fluidic layer was replaced with 1 mL of the CHO cell suspension. The actual number trapped and the distribution map of the trapped cells in the device are provided in the Appendix A. The device was then allowed to stand for 3 min to trap the cells into a microwell layer randomly. Afterwards, approximately 2.8 mL of the mixed suspension was pipetted out from the fluidic layer. Finally, the validation of single cells trapping in the microwell layer was assessed with an inverted microscope (TS100F, Nikon, Melville, NY, USA).

### 2.4. Optimization of the Cell Loading Concentration and Settling Time for MBM Device

To optimize the cell loading concentration, the CHO cell suspensions with different cell concentrations (cell mL^−1^) of 10,000, 12,500, 15,000, and 17,500 were prepared using Luna-II^TM^ automated cell counter (Logos Biosystems, Dongan-gu Anyang-si, Gyeonggi-do, South Korea) with trypan blue exclusion method [19]. The percentage of single cells trapping was calculated as the amount of single-cell trapping in an entire microwell layer. The cell loading concentration, which provides the highest percentage of single-cell trapping, was chosen as an optimum cell loading concentration for the MBM device. 

For optimization of trapping time, the different time points (min) of 1, 5, 7, 10, and 3 min were used with an optimum cell loading concentration. The trapping time, which offers the highest percentage of single-cell trapping, was used as an optimum trapping time for the MBM device.

### 2.5. Limiting Dilution Technique

The protocol of the limiting dilution technique was divided into three parts, including cell preparation, cell loading, and single-cell assessment, in accordance with the operation of the MBM device. In brief, the CHO cells were prepared and serially diluted with sterile PBS to an optimal cell concentration of 4.5 cells mL^−1^ [20]. Then, 200 µL of diluted cell suspension was loaded into 200 wells of 96-well plate using a multichannel pipette (Gilson Pipetman L Multichannel, Middleton, USA). Subsequently, the single-cell assessment of the whole plate was validated and quantified by an inverted microscope (TS100F, Nikon, Melville, NY, USA).

### 2.6. Evaluation of Tester Efficiency of Single-Cell Isolation by Limiting Dilution and MBM Device

The ten testers, who belonged to the experienced group (*n* = 5), and the novice group (*n* = 5), were employed to perform the single-cell isolation for both techniques. All testers had not had any experience with the MBM device. Moreover, only the experienced testers were familiar with the limiting dilution technique. The induction of both techniques, including the protocol and the validation of single-cell isolation, was performed by either or both of the tester groups. All testers were then allowed to practice until assuring full acquisition of both techniques.

The percentage of single-cell trapping and the time spent on each process, including cell loading, cell sorting, and single-cell assessment, were recorded to evaluate the efficiency of both techniques. The sequence of both techniques for the testers was completely randomized, and all testers were allowed to rest between performing each technique to avoid human error.

### 2.7. Correctness of Single-Cell Trapping Identification Obtained from Experienced Tester and Novice Tester

The principal investigator re-evaluated the percentage of single-cell trapping utilizing limiting dilution and MBM device, which was collected from both tester groups (*n* = 10) from the previous experiment. Calculating the obtained single-cell trapping between the testers and the principal investigator allowed for the determination of the correction of single-cell trapping.

### 2.8. Statistical Analysis

All experimental data are reported as mean ± standard deviation for at least three independent replicates of the experiment. The statistical analysis of the data was carried out using one-way analysis of variance (ANOVA) with IBM SPSS Statistics (SPSS Inc., Version 21.0, Chicago, IL, USA). To evaluate the significance of differences between groups, paired *t*-tests and Duncan’s new multiple-range tests were applied. The level of statistical significance was defined as *p* < 0.05.

## 3. Results

### 3.1. Optimization of Cell Loading Concentration and SettlingTime for Microwell-Based Microfluidic Device

The percentage of (wells with) single-cell trapping using the MBM device with different cell loading concentrations of 10,000, 12,500, 15,000, and 17,500 cells mL^−1^ are shown in Figure 3a. The settling time was fixed at 3 min. The percentage of single-cell trapping wells was significantly increased with the cell concentrations from 10,000–15,000 cell mL^−1^ (*p* < 0.05). The highest value was achieved at the cell loading concentration of 15,000 cells mL^−1^ (31.0 ± 3.0%) and dropped at 17,500 cells mL^−1^ (14.5 ± 0.9%). The dropped percentage was not statistically different from the control at the cell loading concentration of 10,000 cells mL^−1^ (12.5 ± 0.5%) (Appendix A). Thus, the cell loading concentration at 15,000 cells mL^−1^ was chosen as the optimal concentration based on the highest percentage of single-cell trapping. For the settling time optimization, the different time points at 1, 3, 5, 7, and 9 min were tested using the optimal cell loading concentration obtained from the above experiment. The percentage of single-cell trapping at different time points is presented in Figure 3b. The highest significantly different single-cell trapping percentage was at the settling time point of 5 min (32.2 ± 0.3%), while a significant decrease in single-cell trapping found was observed at the settling time points of 1 min (10.2 ± 2.4%) and 9 min (4.7 ± 0.3%). However, there was no significant difference at the settling time point of 7 min (17.5 ± 1.8%) compared to the control at 3 min (16.0 ± 1.5%) (Appendix A). Hence, the settling time point at 5 min was then selected as the optimal value according to the cell loading concentration of 15,000 cell mL^−1^ for operating the MBM device. Images of different trapped cells in the MBM device are shown in Figure 3c–e, including no-CHO cell trapping, single-CHO cell trapping, and multi-CHO cell trapping.

### 3.2. Efficiency of Single-Cell Isolation Using Limiting Dilution and the Microwell-Based Microfluidic Device

The efficiency of single-cell isolation was demonstrated by the percentage of single-cell trapping obtained from the limiting dilution and the MBM device. The single-cell trapping, which was performed by the testers (*n* = 10), is shown in Figure 4. It was found that the MBM device offered a significantly higher value of single-cell trapping at 24.3 ± 4.8% (*p* < 0.001) compared to the limiting dilution technique at 5.2 ± 0.8% (Appendix A).

### 3.3. Time Spent on Single-Cell Isolation between Limiting Dilution and MBM Device

The average time spent on single-cell isolation was separated into three main procedures, namely cell preparation, cell loading, and single-cell assessment. The time spent by the 10 testers on each procedure and the total time spent processing are depicted in Figure 5. A significantly lower time spent on cell preparation and single-cell assessment was shown in the MBM device (*p* < 0.001), with values of 2.4 ± 0.1 min and 8.5 ± 3.0 min, respectively. In comparison, a significantly higher time spent (*p* < 0.001) using the MBM device was observed in cell loading, with a value of 7.7 ± 0.6 min, compared to limiting dilution. However, the total time-spent processing of the MBM device significantly exhibited less time spent (*p* < 0.001) at 18.6 ± 3.1 min, compared to limiting dilution at 27.9 ± 8.3 min (Appendix A).

### 3.4. Correctness of Single-Cell Trapping Identification

Single-cell identification has to be performed by the testers as a part of the single-cell isolation. An ability to distinguish the trapped single cells from others (i.e., clumps of multiple cells and debris) can be greatly affected by the optics of and light interference within the cell containers, especially if the testers are not experienced in working with the cells. As a part of an evaluation of the feasibility of this device, we compared the correctness of the single-cell trapping identification by testers of both groups using two different cell isolation methods.

The correctness of obtained single-cell trapping using the limiting dilution technique and the MBM device on the experienced and novice groups is presented in Figure 6. It was found that the MBM device could increase the percentage of correctness on both types of testers, especially and significantly in the novice group (*p* < 0.05) at 89.9 ± 17.7%, compared to limiting dilution at 73.2 ± 16.3% (Appendix A). However, there was no significant difference found between either technique in the experienced testers group.

## 4. Discussion

Cell loading concentration and settling time are critical parameters in obtaining a high percentage of single-cell trapping by the MBM device. The percentage of single-cell trapping was enhanced by increasing the cell loading concentration and the settling time. However, the reduction of single-cell trapping was detected when the highest cell loading concentration and the settling time were further increased. It was also noticed that the cells were randomly trapped into each hole of the microwell layer, resulting in either no cell trapped, a single cell trapped, or multiple cells trapped. The optimal number of cell loading concentrations and the settling time can thus directly affect the rate of single-cell trapping. In contrast, lower and higher values of these parameters can increase the no-cell trapped and multiple-cell trapped situations, respectively. Our results correspond with the findings of Kobel and colleagues, in which using the longer settling time resulted in a lower single-cell trapping percentage in the microwell device due to the increase of multiple cells per well [21]. Based on the relational distribution between the starting cell concentration or the settling duration and the single-cell trapped event resulting in their optimal values, this observation can be described by Poisson’s distribution [22]. Our finding, therefore, was in agreement with Kuntanawat and colleagues that the highest single-cell trapping is related to the optimal initial cell concentration [16]. It has been recognized that the geometry of the wells and cell size and type can contribute to the efficiency of single-cell isolation [16,23]. In this particular study, the well size of 1000 × 1000 × 1000 µm was chosen because it was demonstrated to be capable of trapping cells of different sizes and shapes efficiently [16]. Optimization of the wells’ geometry with regard to different cell types/sizes may help improve the single-cell efficiency.

The efficiency of single-cell isolation between the limiting dilution technique and the MBM device was investigated by ten testers belonging either to the experienced group or the novice group. A significant 4.3-folded increase in single-cell trapping was noticed in the MBM device technique group. This finding could be explained by the optimal number of cell-loading concentrations for the MBM device (15,000 cells mL^−1^) and the total number of actual cells trapped (Appendix A), which is higher than the optimal number for the cell-loading concentration of the limiting dilution technique (4.5 cells mL^−1^) [24]. Using the MBM device significantly showed decreased time for total time-spent processing to 30 min compared to the limiting dilution technique.

This finding can be explained by shortening the time for cell preparation and the single-cell assessment process. The one-step dilution is a crucial factor of the MBM device, allowing a 2.63-fold decrease in time during the cell dilution process compared to the limiting dilution technique. Indeed, as the number of optimal initial cell concentrations (15,000 cell mL^−1^) of the MBM device is already high, a single dilution can be performed from the initial cell mixture, which usually has a high cell concentration of 10^6^ cells mL^−1^ for this final cell concentration. However, a large number of serial dilutions was required in the limiting dilution technique due to its comparatively very low optimal initial cell concentration number (4.5 cell mL^−1^) [22], which consequently increased the time spent in the process. Moreover, the MBM device demonstrated a significant 2.13-fold decrease in time spent during single-cell assessment compared to the limiting dilution technique. This finding could be explained by the lower observation area in each well of the MBM device (1.00 mm^2^) in comparison to each well of the 96-well plate (43.58 mm^2^). Therefore, the effortless single-cell investigation could be attained in the MBM device due to the higher ratio between CHO cell size and the observation area.

Even though the time spent during the cell loading process of the MBM device (7.7 min) was higher than the limiting dilution technique (3.5 min), it should be noted that the increased time resulted from a 5 min wait during the cell trapping process. Hence, the actual time spent for cell loading is 2.3 min, shorter than the actual operating time of the limiting dilution technique. This finding could be explained by the one-step pipetting of cell suspension into the MBM device, which provided a less-laborious option compared to the limiting dilution technique, where a larger amount of pipetting was required to fill the 200 wells of 96-well plates. Furthermore, the MBM device demonstrated a higher percentage of single-cell trapping correctness than the limiting dilution technique, especially significant in the novice group. This finding indicates that the MBM device could improve the accuracy of single-cell observations reflected by effortless single-cell investigation under an inverted microscope, as mentioned before.

Additionally, the MBM device not only presents a higher single-cell isolation efficiency with a lower total time spent, reagent consumption, and skill intensity than the conventional technique, i.e., the limiting dilution (Table 1), it can also offer an alternative approach to more high-end approaches, such as fluorescence-activated cell sorting (FACS) (Table 1). Moreover, even though the MBM device can prove to be less efficient than the FACS technique, it nonetheless offers a lower consumption of reagents and no need for highly skilled performers at lower associated costs. Compared to most of the microfluidic devices of a similar kind, the advantages of an MBM device are low production cost, reusability, and minimal requirement of associated technical equipment to operate. Microfluidic devices can be complicated to fabricate [13,14] and are not always intended for multiple uses, which results in elevated costs in production. The MBM device, on the other hand, was made of a very affordable material (Poly methyl methacrylate) and fabricated using a fast and cost-effective method (laser cutting). It was demonstrated in our previous work [25] that a former prototype of the MBM device can be reused after being disinfected with 70% (*v*/*v*) ethanol and autoclaving for at least up to 10 times with no significant drop in trapping efficiency. Unlike most devices invented for a similar purpose that many cases require either specialized microscopic, optics, or automation equipment to properly operate [13,15], a simple bright-field inverted microscope is sufficient to perform cell isolation using our technique.

Hence, this device could serve as a bench-top approach as a simple, affordable, efficient, and less skill-intensive alternative to the conventional technique of single-cell isolation and remains accessible to general laboratories for monoclonal cell line establishment.

## 5. Conclusions

We successfully implemented and optimized a bench-top MBM approach for single-cell isolation of monoclonal cell lines. The MBM device not only significantly improved the efficiency of single-cell isolation in comparison to the conventional technique of limiting dilution, but it also provided a lower consumption of resources coupled with an effortless operating modus. Furthermore, this device proved to be a simple, affordable technique and can be implemented by general laboratories as an alternative approach for monoclonal cell line production.

## Figures and Tables

**Figure 1 micromachines-13-01939-f001:**
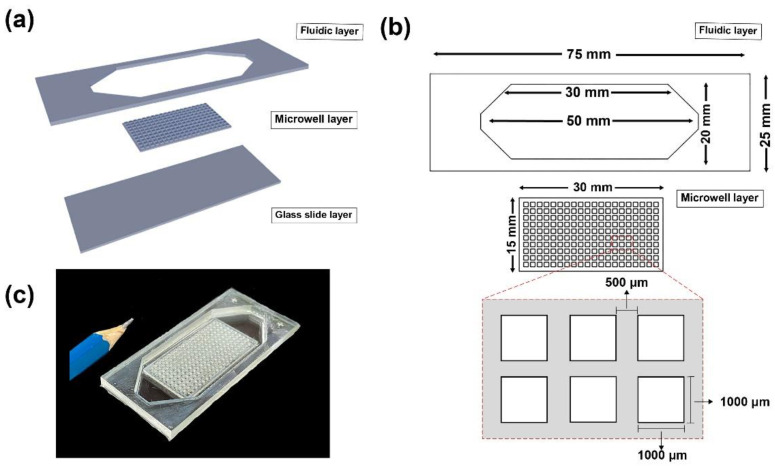
The microwell-based microfluidic (MBM) device for single-cell isolation. (**a**) The component of the MBM device includes a fluidic layer, a microwell layer, and a glass slide layer. (**b**) The details of the MBM device. (**c**) Image of the fabricated MBM device.

**Figure 2 micromachines-13-01939-f002:**
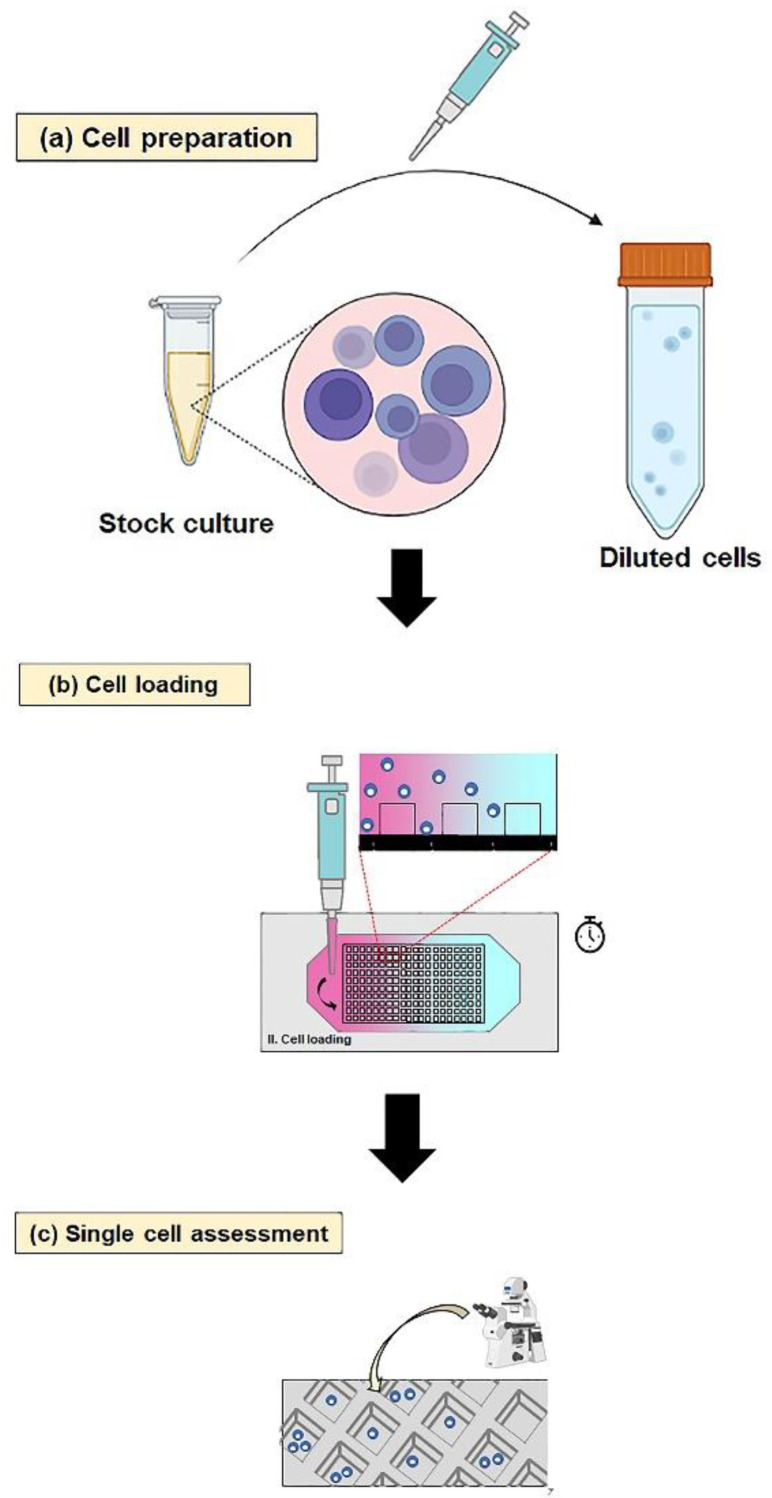
The process of operation for single-cell isolation using the MBM device. (**a**) Cell preparation of initial cell concentration by dilution method, (**b**) cell culture medium and cell loading into the MBM device, (**c**) single-cell assessment under an inverted microscope using 10× magnification.

**Figure 3 micromachines-13-01939-f003:**
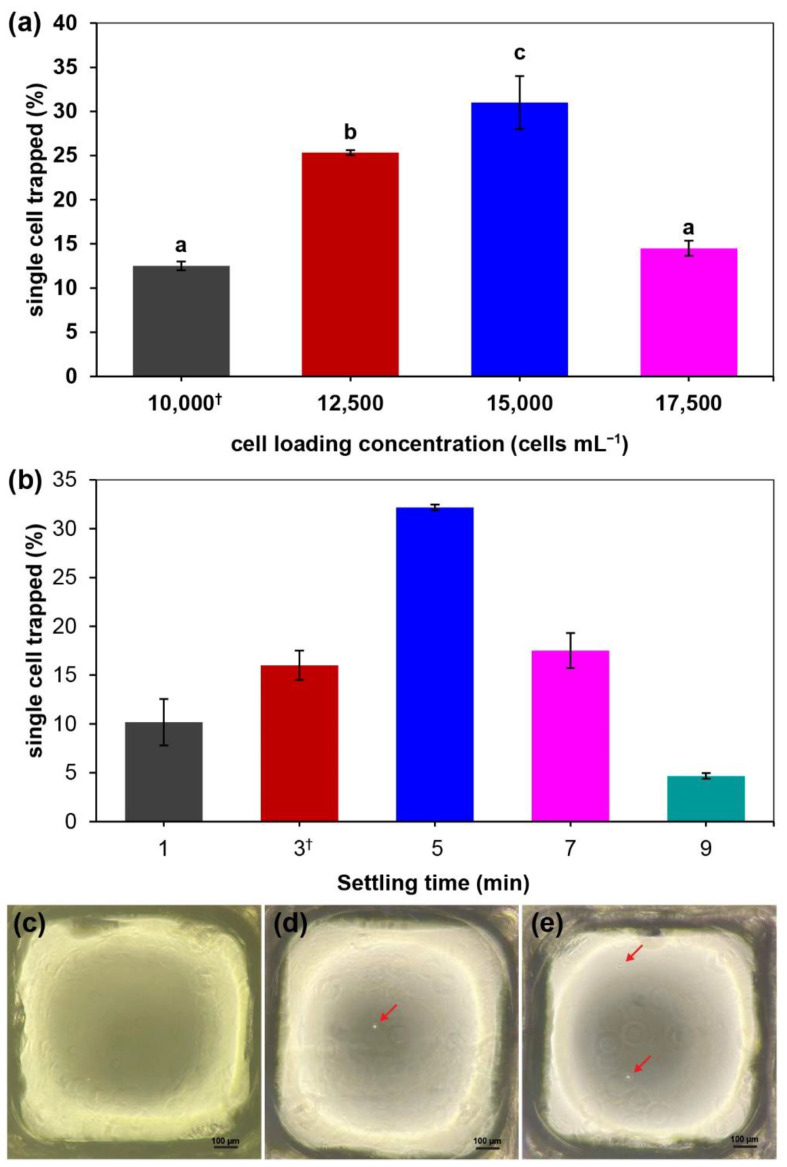
Average single-cell trapping using the MBM microfluidic device with optimization of (**a**) cell loading concentration (cell mL^−1^) and (**b**) the settling time (min). Images of the MBM’s wells with (**c**) no-CHO cell trapping, (**d**) a single-CHO cell trapping, and (**e**) multi-CHO cell trapping. Red arrow indicates CHO cells. Scale bars represent 100 µm. Experimental data are reported as mean ± standard deviation (*n* = 3), with † corresponding to the control. Means followed by the different letters within columns indicate a significant difference at *p* < 0.05 using Duncan’s multiple range test.

**Figure 4 micromachines-13-01939-f004:**
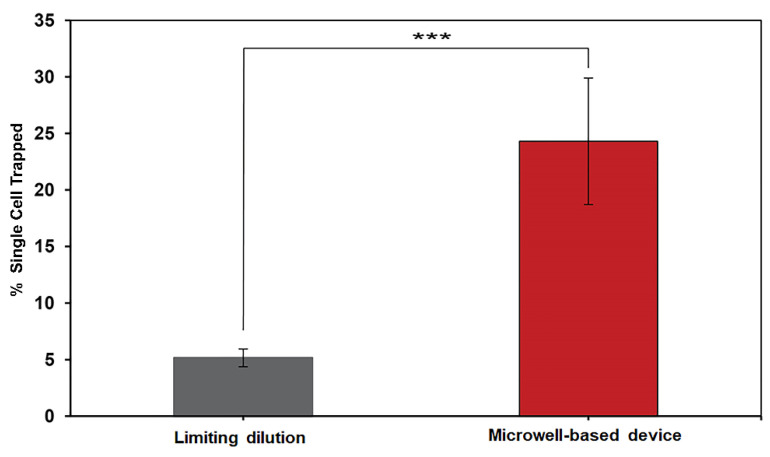
Comparison of the percentage of single-cell trapping using limiting dilution technique and the MBM device. Experimental data are reported as mean ± standard deviation (*n* = 10). Means followed by the asterisk indicate a significant difference (*p* < 0.001) using paired *t*-test.

**Figure 5 micromachines-13-01939-f005:**
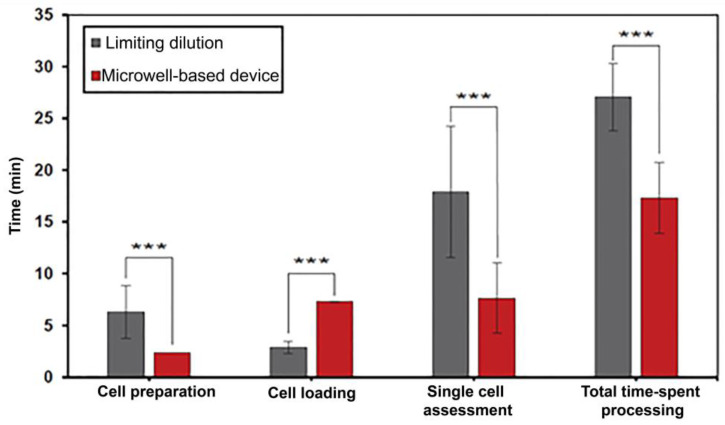
The time-spending using limiting dilution and a microwell-based microfluidic device. The data were based on each category, including preparation, cell loading, single-cell assessment, and total time-spent processing. Experimental data are reported as mean ± standard deviation (*n* = 10). Means followed by the asterisk within columns indicate a significant difference at *p* < 0.001 using paired *t*-test. Each column was analyzed separately.

**Figure 6 micromachines-13-01939-f006:**
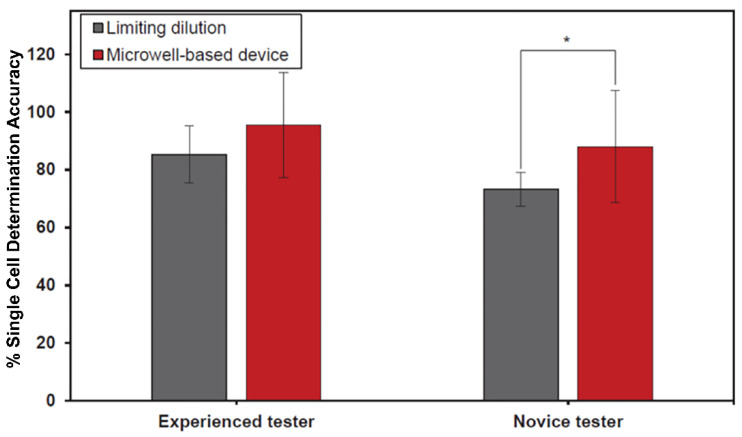
Comparison of the percentage of single-cell trapping correctness using limiting dilution and the MBM device. The results obtained from the experienced group (*n* = 5) and the novice group (*n* = 5) were reported as mean ± standard deviation. The asterisk between columns indicates a significant difference at *p* < 0.05 using a paired *t*-test. Each pair of columns was analyzed separately.

**Table 1 micromachines-13-01939-t001:** Comparison of single-cell cloning using the MBM device, the limiting dilution technique, and the Fluorescence-Activated Cell Sorting (FACS) technique.

Parameters	Limiting DilutionTechnique	Microwell-Based Microfluidic Device	Fluorescence-Activated Cell Sorting (FACS) Technique [26]
Total time spent ^a^	High	Low	Low
Reagent consumption	High	Low	High
Skill intensity	Moderate	Low	High
Single-cell isolation efficiency ^b^	Low	Moderate	High

^a^ Total time spent refers to the time spent during the whole single-cell isolation operation, including cell preparation, cell loading/isolation, and single-cell assessment. ^b^ Single-cell isolation efficiency represents the monoclonality of the isolated cells obtained from certain techniques.

## Data Availability

The data presented in this study are available on request from the corresponding author.

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
