# Peer review of "A Bench-Top Approach for Isolation of Single Antibody Producing Chinese Hamster Ovary (CHO) Cells Using a Microwell-Based Microfluidic Device"

_micromachines, 2022, doi:10.3390/mi13111939_

Round 1
Reviewer 1 Report
This manuscript describes a bench-top approach for single cell isolation using microwell-based microfluidic device, and compared it with conventional methods for single cell cloning including limited dilution and fluorescence-activated cell sorting. Although this developed method is attractive in its simplicity, some questions about its novelty and data need to be addressed.
1) The concentration of CHO cell suspension is not clear. The concept of microwell-based microfluidic device is similar to limiting dilution, which is a traditional method used for single cell cloning. It is exclusively based on statistical probabilities for single cell isolation, which depend on the average number of cells seeded per well according to the Poisson distribution. In the manuscript, the CHO cell suspension was prepared to approximate a cell concentration of 10,000 cells/mL, and 1 mL replaced the culture medium. Afterwards, 2.8 mL of the mixed suspension was pipetted out. The actual cell number loaded into the device is not clear.
2) As for the comparison with the limiting dilution method, most of the literature reported the monoclonality or single cell efficiency is at least around 10% or above (regular range around 10-30%). In this manuscript, according to Fig 4, the single cell trapping efficiency is only around 5%, which is much lower than expected. In the microwell-based device, the single cell trapping efficiency is around 25% while the fluorescence-activated cell sorting even can reach 60%. Thus it is not clear how the microwell-based method improves single cell trapping efficiency in the literature context.
3) Have the authors compared the cell isolation efficiency between assembled regular glass slide and positive charge microscope slide?
4) Can the authors comment on if the cell size or cell type will affect the isolation efficienc?
5) In the methods part 2.4, the authors mentioned: “the CHO cell suspension with different cell concentrations (cell / mL) of 12,500, 15,000, 17,500, and 10,000 were prepared using Luna-IITM automated cell counter.” The different cell concentrations should be listed in a reasonable order.
Author Response
This manuscript describes a bench-top approach for single cell isolation using microwell-based microfluidic device, and compared it with conventional methods for single cell cloning including limited dilution and fluorescence-activated cell sorting. Although this developed method is attractive in its simplicity, some questions about its novelty and data need to be addressed.
- The concentration of CHO cell suspension is not clear. The concept of microwell-based microfluidic device is similar to limiting dilution, which is a traditional method used for single cell cloning. It is exclusively based on statistical probabilities for single cell isolation, which depend on the average number of cells seeded per well according to the Poisson distribution.
In the manuscript, the CHO cell suspension was prepared to approximate a cell concentration of 10,000 cells/mL, and 1 mL replaced the culture medium. Afterwards, 2.8 mL of the mixed suspension was pipetted out. The actual cell number loaded into the device is not clear.
Response: The actual cells loaded to the device, in this case, was 10,000 cells because 1 ml of the 10,000 cells/ml cell suspension was loaded. However not all the cells were trapped and were later flushed out of the device. We decided to express the amount of the cells loaded into the device as a volume of cell suspension at a specific initial concentration instead of the actual cell used for the ease of cell preparation. We appreciate you pointing out this point as it shows that our description on this matter could cause the confusion. For this reason, we elaborate the cell loading protocol in section 2.3 (lines 136-138) and provide the trapped cell number and distribution maps of the cells trapped in the device when different cell loading concentrations were used in the supplementary material S1.
2- As for the comparison with the limiting dilution method, most of the literature reported the monoclonality or single cell efficiency is at least around 10% or above (regular range around 10-30%). In this manuscript, according to Fig 4, the single cell trapping efficiency is only around 5%, which is much lower than expected. In the microwell-based device, the single cell trapping efficiency is around 25% while the fluorescence-activated cell sorting even can reach 60%. Thus, it is not clear how the microwell-based method improves single cell trapping efficiency in the literature context.
Response:
5% single cell efficiency of our limiting dilution experiment were obtained from testers with different level of skill and experience. As mentioned in the experimental section (2.6), we intentionally included less experienced and skilled testers in the experiment to showcase that our invented technique can be performed with less demand for skill and technical experience. This might explain why the single cell efficiency is much lower than those reported by others.
The MBM device may not be the best technique when it comes to the single cell efficiency alone. However, it was demonstrated to be a middle ground between the conventional technique and the high-end methodology. Comparing the conventional limiting dilution method, the MBM technique offered a moderate single cell efficiency but less time-consuming and less reagent- and skill-intensive. Although MBM device does not give as high single cell efficiency as FACS, it does not require access to high-end equipment, highly skilled performers and large quantities of reagent.
Comparing to most of the microfluidic devices of its kind, the advantages of our device are low production cost, reusability and minimal requirement of associated technical equipment to operate. Most of the microfluidic devices are complicated to fabricate [15, 16] and not usually reusable which result elevated costs in the production. Our device was made of a very affordable material (Poly methyl methacrylate) and fabricated using a fast and cost-effective method (laser cutting). We have also demonstrated in our previous work that a similar device can be reused after cleaning and autoclaving for up to at least 10 times [25]. Unlike most devices invented for a similar proposed which in many cases require either specialized microscopic, optics or automation equipment to properly operate [15, 16, 17], a simple bright field inverted microscope is sufficient to perform cell isolation using our technique.
We have elaborate these points in multiple locations of our revised manuscript (Lines: 70-74, 84-86, 345-357)
3- Have the authors compared the cell isolation efficiency between assembled regular glass slide and positive charge microscope slide?
Response: Done. To clarify this point, we have added the mechanism of the MBM device into the text. We have not attempted the cell isolation using a regular glass slide in this work. It was demonstrated in our previous work [18] to be ineffective. Cells can be easily escaped from the wells when the electrostatic interaction between the positively charged glass slide and the negatively charged cell surface is absent. We added this point in Lines 110-113.
4- Can the authors comment on if the cell size or cell type will affect the isolation efficiency?
Response: Done. The cell size and types and the wells’ geometry are known to affect the isolation efficiency. We elaborated on this point, in lines 290-295.
5- In the methods part 2.4, the authors mentioned: “the CHO cell suspension with different cell concentrations (cell / mL) of 12,500, 15,000, 17,500, and 10,000 were prepared using Luna-IITM automated cell counter.” The different cell concentrations should be listed in a reasonable order.
Response: Done. The different cell concentrations have been listed in a reasonable order: 10,000, 12,500, 15,000, and 17,500. (Line 149)

Reviewer 2 Report
In this manuscript, the authors reported a simple microwell-based microfluidic device for isolating single high-antibody producing CHO cells. The isolation parameters of cell loading concentration and settling time were optimized, and the general limiting dilution method was used to demonstrate the merits of the reported microfluidic device. Although simple and fast single high-antibody producing CHO cell isolation has been achieved in the work, the novelty of the proposed method is limited as it is not difficult to find that similar papers have been published before (Biosensors 12.2 (2022): 58; PloS one 10.11 (2015): e0139980). Hence, in my opinion, the manuscript is not ready for publication.
1. The novelty of this method is not well-documented. No performance comparation of this method and previous methods are given. what's the main advantage of this method compared to others?
2. The total number of microwell is 200. Each well had a dimension (width x length x depth) of 1000 × 1000 x 1000 µm. Why do the authors select these parameter for this study? Is there any effect of total microwell number and well dimension on the efficiency of single cell isolation?
3. It’s better to add cell or device images for the presented data shown in Figure 3-6.
4. Please give more information for the labeling of ‘a, b, c, a’in Figure 3a & b.
5. For the table 1, it’s better to specify the ‘Total time spent’, ‘Single-cell isolation efficiency’to show the merits of the proposed method.
Author Response
Reviewer 2
In this manuscript, the authors reported a simple microwell-based microfluidic device for isolating single high-antibody producing CHO cells. The isolation parameters of cell loading concentration and settling time were optimized, and the general limiting dilution method was used to demonstrate the merits of the reported microfluidic device. Although simple and fast single high-antibody producing CHO cell isolation has been achieved in the work, the novelty of the proposed method is limited as it is not difficult to find that similar papers have been published before (Biosensors 12.2 (2022): 58; PloS one 10.11 (2015): e0139980). Hence, in my opinion, the manuscript is not ready for publication:
- The novelty of this method is not well-documented. No performance comparation of this method and previous methods are given. what's the main advantage of this method compared to others?
Response: We do agree that there are countless number of papers employing microwell mechanisms for different applications as can be seen in the publications suggested by Reviewer 2. However, the idea of implementing the modified version of our previously published work in isolation of the antibody-producing cells is novel. The technique offers a new alternative in the isolation of antibody-producing cells for linage establishment that is ready to be implemented in general cell laboratories dealing with establishment of CHO cells expressing potential therapeutic antibody.
We’ve demonstrated that our newly proposed concept can serve as a middle ground between the conventional technique and the high-end methodology. Comparing the conventional limiting dilution method, our invented technique offered a moderate single cell efficiency but was less time-consuming and less reagent- and skill-intensive. Although the device did not give as high single cell efficiency as FACS, it does not require access to high-end equipment, highly skilled performers and large quantities of reagent.
Comparing to most of the microfluidic devices of its kind, the advantages of our device are low production cost, reusability and minimal requirement of associated technical equipment to operate. Most of the microfluidic devices are complicated to fabricate [15, 16] and not usually reusable which results elevated costs in the production. Our device was made of a very affordable material (Poly methyl methacrylate) and fabricated using a fast and cost-effective method (laser cutting). We have also demonstrated in our previous work that a similar device can be reused after cleaning and autoclaving for up to at least 10 times [25]. Unlike most devices invented for a similar proposed which in many cases require either specialized microscopic, optics or automation equipment to properly operate [15, 16, 17], a simple bright field inverted microscope is sufficient to perform cell isolation using our technique.
We have elaborated these points in multiple locations in the revised manuscript (Lines: 70-74, 84-86, 345-357).
2- The total number of microwell is 200. Each well had a dimension (width x length x depth) of 1000 × 1000 x 1000 µm. Why do the authors select these parameters for this study? Is there any effect of total microwell number and well dimension on the efficiency of single cell isolation?
Response: The reason is that it was found this well’s dimension is appropriate for trapping single cells of different organisms with different shapes and sizes in our previous work [18]. In addition, it fits well with the size of the micropipette tip which also allows for the manual transfer of the cell(s) right under the microscope using a micropipette. There is a limited working area where cells can be deposited in the device, so the smaller the well size, the greater number of the wells can fit it. The well dimension can affect the trapping efficiency [18]. We believe that the well dimension optimization could be a topic of our further study. We have elaborated on these points in lines 290-295.
3- It’s better to add cell or device images for the presented data shown in Figure 3-6.
Response: Done. Images of the trapped cell(s) in the MBM device are now added to the revised manuscript. This includes the wells of single-cell trapped, multi-cell trapped, and no-cell trapped (Line: 220).
4- Please give more information for the labeling of ‘a, b, c, a’in Figure 3a & b.
Response: Done. The labelling of letters (a, b, c, d) within columns in Figures 3a and 3b indicate a significant difference at p < 0.05 using Duncan’s multiple range test. To clarify readers, we have added more information to the text (Lines: 222-226).
5- For the table 1, it’s better to specify the ‘Total time spent’, ‘Single-cell isolation efficiency’ to show the merits of the proposed method.
Response: Done. (Lines: 334-337) the authors have added the specific definition of the total time spent and single-cell isolation in the table legend.

Round 2
Reviewer 1 Report
The authors have addressed my previous comments.
Reviewer 2 Report
I have no further comments.